# From Plankton to Primates: How VSP Sequence Diversity Shapes Voltage Sensing

**DOI:** 10.3390/ijms262210963

**Published:** 2025-11-12

**Authors:** Lee Min Leong, Youna Kim, Bradley J. Baker

**Affiliations:** 1Brain Science Institute, Korea Institute of Science and Technology, Seoul 02792, Republic of Korea; 2Department of Biological Science, Florida State University, Tallahassee, FL 32306, USA; 3Division of Bio-Medical Science and Technology, KIST School, Korea University of Science and Technology (UST), Seoul 02792, Republic of Korea

**Keywords:** voltage-sensing phosphatase (VSP), voltage-sensing domain (VSD), genetically encoded voltage indicator (GEVI)

## Abstract

Voltage-sensing phosphatases (VSPs) provide a conserved framework for dissecting the mechanics of voltage sensing and for engineering genetically encoded voltage indicators (GEVIs). To evaluate how natural sequence diversity shapes function, we compared VSP voltage-sensing domains (VSDs) from multiple species by replacing the phosphatase domain with a fluorescent protein to enable optical detection of VSD responses. Every construct that reached the plasma membrane produced a voltage-dependent optical signal, underscoring the deep conservation of voltage sensing across VSP orthologs. Yet lineage-specific substitutions generated strikingly different phenotypes. A plankton VSP ortholog from *Eurytemora carolleeae* and the Sea Hare (*Aplysia californica*) VSP exhibited left-shifted activation ranges, producing robust fluorescence transitions during modest depolarizations of the plasma membrane. The human VSD of hVSP2 yielded weak, sluggish responses with poor recovery, but reintroduction of a conserved arginine in S1 (G95R) partially restored reversibility, implicating lipid-facing residues in conformational stability. The Chinese hamster (*Cricetulus griseus*) VSD, with atypical S4 sensing charges (RWIR), generated a slow fluorescence increase during depolarization, while reverting to the consensus arginine (RRIR) inverted the polarity to a decrease. These contrasting behaviors show that single residue changes can reshape how VSD movements influence the fluorescent reporter, highlighting the molecular precision revealed by GEVI measurements. Together, these results show that voltage-dependent signaling is deeply conserved across VSPs but shaped by lineage-specific sequence variation, establishing VSPs as powerful models for probing voltage sensing and guiding GEVI design.

## 1. Introduction

Genetically encoded voltage indicators (GEVIs) provide a means to optically monitor membrane potential changes in living cells [1,2]. Among them, the ArcLight family [3,4,5] and its derivatives use the voltage-sensing domain (VSD) from the *Ciona intestinalis* [6] voltage-sensing phosphatase (VSP) gene fused to a fluorescent protein producing robust voltage-dependent changes in fluorescence. While initially developed as reporters of electrical activity, these constructs also provide a powerful framework for investigating how sequence variation in VSDs influences conformational dynamics and their coupling to optical signals [7]. Because different neuronal processes operate across distinct voltage ranges, the ability to tune the voltage dependence of GEVI responses is an important goal for extending their utility [8].

VSP orthologs display remarkable sequence diversity across species, particularly in the distribution of positively charged residues within the S4 helix of the VSD [7,9]. These sensing charges drive transmembrane movement of S4 in response to changes in voltage, and their number and positioning vary across orthologs. Despite this diversity, little is known about how naturally occurring VSP sequences map onto the voltage dependence and fluorescence responses of ArcLight-type GEVI constructs.

Here, we demonstrate that multiple VSP orthologs and variants can produce voltage-dependent optical signals when expressed in the ArcLight framework. Humans have two versions of VSP, hVSP1 and hVSP2. Human VSD (hVSP2 [10]) traffics efficiently to the plasma membrane and yields responses within the physiological voltage range, as does mouse, though mouse exhibits poorer membrane localization. Remarkably, the Chinese hamster variant, which retains only two potential sensing charges in S4 (positive charges in S4 responsible for voltage-dependent conformational changes), still supports detectable movement in response to 100 mV depolarization. Finally, introducing an additional sensing charge by mutating the hamster W→R at position R2 inverts the voltage-dependent fluorescence response, indicating that the fluorescent protein environment shifts. This suggests that altered S4 interactions with countercharges in S1–S3 reorient the cytoplasmic fluorescent domain, producing a fundamentally different optical output.

Together, these results establish that natural variation in VSP sequences can be exploited to reveal how sensing charge number and distribution shape VSD movement and the subsequent fluorescence response. Beyond demonstrating that human and other orthologs function in the physiological range, this comparative approach shows that even unusual variants, such as hamster, uncover new principles of ArcLight-type GEVI design and mechanism.

## 2. Results

### 2.1. Exploring Sequence Diversity in the VSD of the VSP Gene Family

The first GEVI to function in mammalian cells was VSFP2 [11] which incorporated the VSD from *Ciona intestinalis* (Sea squirt) VSP. Although these initial constructs trafficked efficiently to the plasma membrane, they required strong depolarizations of the membrane to elicit optical signals. A single substitution in the S4 helix (R217Q—numbering is based on the *Ciona* sequence unless otherwise stated) shifted the voltage dependence into the physiological range [11,12] a breakthrough that enabled crystallographic determination of the *Ciona* VSD in both the resting ‘down’ (R217) and activated ‘up’ (R217E) states [13]. These structures form the foundation for comparative analysis of VSP sequence diversity (Figure 1A).

To investigate how natural sequence variation influences voltage sensing, we aligned 377 unique VSP VSD sequences spanning plankton, fish, reptiles, birds, arachnids, and mammals. Sequence logos [15] of the four transmembrane helices revealed clusters of conserved residues, often positioned along one face of the helix, consistent with stabilizing electrostatic interactions within the VSD (Figure 1B). Notably, *Ciona* contains several deviations from consensus residues (e.g., S154A in S2—the consensus sequence is serine, the *Ciona* sequence is alanine), high-lighting sites where evolutionary substitutions may alter voltage-dependent function.

Previous mutagenesis studies at some of these conserved positions modified GEVI response amplitudes, kinetics, or voltage ranges [7]. However, targeted mutations cannot fully reproduce the broader stabilizing context encoded by naturally evolved variants. To directly test how evolutionary diversity shapes VSD behavior, we synthesized several representative ortholog constructs based on intriguing sequence divergence. Those presented in this study are listed in Appendix A.

For experimental characterization, the catalytic phosphatase domain of each VSP was replaced with the fluorescent protein Super Ecliptic pHluorin A227D, leaving the intact VSD and its linker region. This design (Figure 1C) allowed us to evaluate plasma membrane localization and to measure voltage-dependent optical responses across orthologs.

### 2.2. Expression and Trafficking of VSP–FP Chimeras Were Highly Variable Across Species

We next examined the cellular distribution of the ortholog constructs in HEK293 cells. Constructs showed striking differences in their ability to reach the plasma membrane. Several orthologs, including *Ciona*, human (hVSP2), and finch, displayed obvious plasma membrane localization with minimal intracellular accumulation (Figure 2A). In contrast, others, such as hamster, honey bee mite, and mouse, showed poor membrane targeting and instead accumulated within intracellular compartments (Figure 2B).

These results highlight that not all VSP orthologs are well expressed in this mammalian system. In particular, the poor membrane localization of spider and honey bee mite, and even the reduced membrane targeting of mouse, may reflect limitations of using HEK 293 cells rather than strict evolutionary differences in trafficking regulation. Thus, while some orthologs readily localize to the plasma membrane in this system, others may require more compatible host environments to reveal their true trafficking potential.

All constructs that trafficked efficiently to the plasma membrane yielded a voltage-dependent optical signal, although several required strong depolarizations to elicit responses (Table 1). The sea hare and plankton VSD orthologs exhibited the most left-shifted response, producing a clear signal with only a 100 mV depolarization (Figure 1C). By contrast, *Takifugu*, finch, rice fish, and sea turtle required very strong depolarization of the plasma membrane to generate detectable responses. (Note: the membrane resistance of HEK cells may not allow whole-cell recordings to clamp accurately at +230 mV; these steps were applied simply to depolarize the membrane as much as possible).

Two constructs stood out as exceptions. The plankton VSD generated robust signals at relatively modest depolarization, and the human VSD (hVSP2), despite its weak overall response, produced a reproducible voltage-dependent signal at 100 mV. Unlike other orthologs, the human VSD displayed a unique kinetic profile: its initial fluorescence change was very slow, with a τ_on_ exceeding 100 ms, and the response failed to recover when the membrane potential was repolarized, precluding calculation of τ_off_. Comparison of signal size and kinetics across functional constructs is summarized in Table 1.

Together, these results establish that while many VSP orthologs can generate voltage-dependent signals when expressed in HEK 293 cells, the human VSD represents a surprising outlier, motivating a closer investigation of its unusual sequence features and biophysical behavior.

### 2.3. The VSD Is Responsible for the Voltage-Dependent Optical Signal

These new species-derived GEVIs contain both the N-terminal cytoplasmic sequence and the VSD, raising the possibility that the unusual response of the human construct could be influenced by its N-terminal region rather than by the VSD itself. Notably, AlphaFold [14] predictions suggested that the human sequence contains an elongated S0 helix, whereas the finch (*Lonchura striata*) sequence has a substantially shorter S0 (Figure 3). To test whether differences in this region contributed to the altered optical responses, we generated two chimeras with the fusion site located within the S0 helix.

The Finch/Human chimera, which contains the human VSD (hVSP2), produced a voltage-dependent signal highly similar to that of the original human construct (Figure 3). The signal amplitude was slightly larger, possibly reflecting more efficient trafficking to the plasma membrane and thus an increased number of responsive proteins. Conversely, the Human/Finch chimera, which contains the finch VSD, displayed a response pattern closely matching that of the original finch construct. Importantly, the frame-subtraction image shows that the Human VSD (hVSP2) for both constructs in Figure 3 are restricted to the plasma membrane.

Together, these results demonstrate that the distinct response of the human GEVI is determined primarily by its VSD, with little to no contribution from the N-terminal or S0 sequences. If the S0 helix influences voltage-dependent signaling, its effect is difficult to discern.

### 2.4. Reverting a Primate-Specific Mutation in S1 Partially Recovers the Repolarization Signal in the Human VSD

The voltage-dependent signal of the human VSD (hVSP2) is unusual in that fluorescence does not return to baseline upon repolarization. One potential explanation is a primate-specific mutation in the S1 helix (Figure 4A). In the consensus sequence, position 119 is occupied by an arginine, whereas in primates—including humans—this position is a glycine (G95 in the human sequence).

To test whether this substitution contributes to the atypical response, we reverted the human S1 sequence to the consensus by introducing the G95R mutation. This change improved both the signal size and onset speed of the human VSD voltage response (Figure 4B). For a 50 mV depolarization, the response increased to 1.1% ΔF/F with a τ_on_ of 82.5 ± 3.0 ms. At 100 mV, the signal reached 1.4% ΔF/F with a faster τ_on_ of 41.0 ± 0.7 ms. Although the signal size did not increase further at 150 mV and 200 mV depolarizations, the kinetics improved substantially, with τ_on_ values of 21.5 ± 1.0 ms and 6.3 ± 0.7 ms, respectively.

The G95R mutant also showed partial recovery during repolarization, unlike the wildtype human construct. The τ_off_ was 45.0 ± 1.1 ms for the 50 mV step, slowing to 62.8 ± 2.5 ms at 100 mV and 64.5 ± 6.5 ms at 150 mV. Remarkably, the mutant even produced a detectable response to a 50 mV hyperpolarization step. However, the fluorescence never fully returned to baseline upon return to the holding potential, suggesting that the conformational change is not entirely reversible. Given the position of this residue in S1 (Figure 1B), this behavior may reflect altered interactions with plasma membrane lipids that stabilize different conformational states.

### 2.5. A Divergent S4 Sequence with Only Two Sensing Charges Was Still Capable of Yielding a Voltage-Dependent Signal

All of the non-responding constructs exhibited high internal fluorescence, consistent with poor trafficking to the plasma membrane. To test whether a voltage response could be rescued, we replaced the N-terminus of the Chinese hamster construct with the human sequence, which traffics efficiently. Chinese hamster was chosen for this experiment because it represented the strongest predicted non-responder, owing to its unusual S4 composition.

In most VSP orthologs, the S4 helix contains a positively charged residue every third position, enabling voltage-driven transmembrane movement. While the canonical *Ciona* sequence retains four arginines, most family members follow an R–R–I–R pattern. Hamster is an outlier, with an S4 sequence of R–W–I–R that replaces the second arginine with tryptophan, leaving only two potential sensing charges spaced widely apart. This substitution was expected to both reduce electrostatic drive and sterically hinder S4 movement, and thus we initially hypothesized that the lack of a signal reflected a nonfunctional S4. However, its poor membrane localization needed to be addressed before this prediction could be tested rigorously.

Improved trafficking revealed that the hamster VSD is still capable of generating a voltage-dependent signal. The wildtype hamster construct displayed predominantly intracellular fluorescence and no optical response (Figure 5A). By contrast, when its N-terminus was replaced with the human sequence, surface expression, though still poor, improved sufficiently to unmask a voltage-dependent fluorescence signal (Figure 5B. Further, introducing a W→R substitution at the second S4 position in this human/hamster chimera altered the polarity of the optical response, demonstrating that S4 composition modulates how VSD movements are transduced to the fused fluorescent protein.

Together, these results show that the absence of activity in the wildtype hamster construct was due to poor membrane trafficking rather than an immobile S4. They also support the idea that the unusual hamster S4 sequence influences the positioning and/or movement of the cytosolic FP domain, which underlies the observed optical signal. We have recently shown that altering the chromophore flexibility in the FP domain can result in different fluorescence transition patterns [16]. The VSD from *Ciona* VSP has been shown to dimerize [17] suggesting a potential interaction of neighboring FP domains enabling the development of intermolecular FRET GEVIs [18]. Slight reorganization of FP domain interactions either by altered starting positions and/or altered S4 movement could potentially explain the different polarities of the hamster-based VSD responses.

## 3. Discussion

This study expands the functional analysis of voltage-sensing phosphatase (VSP)-family VSDs by comparing constructs from diverse species. Sequence alignment of 377 orthologs revealed strongly conserved charge–charge interaction patterns across helices S1–S4 yet also highlighted lineage-specific substitutions that alter voltage sensing (Figure 1). Our results show that while the capacity to generate voltage-dependent optical signals is broadly conserved, differences in expression, kinetics, and voltage range may arise from both evolutionary divergence and host–cell-specific factors that influence membrane targeting; however, the contribution of host–cell context remains to be directly tested (Figure 2).

A recurring challenge in this survey was trafficking efficiency in HEK 293 cells. Several orthologs accumulated intracellularly and failed to yield signals, consistent with earlier reports of poor membrane localization for certain VSP family members in mammalian systems [18]. Rescue experiments via N-terminal chimeras confirmed that absence of a signal does not necessarily reflect a nonfunctional VSD (Figure 5). For example, the Chinese hamster construct failed in its wildtype form, but replacement of the N-terminal region with the human sequence (hVSP2) improved membrane localization and unmasked a voltage-dependent optical response. Thus, species-specific differences in trafficking can obscure otherwise functional voltage sensors when expressed in heterologous systems. Indeed, a recent report demonstrated improved membrane localization of mouse VSP via co-expression of basigin [19].

Once located at the plasma membrane, the hamster VSD revealed an unusual optical phenotype distinct from all other species. Depolarization produced a slow, modest fluorescence increase, in contrast to the rapid decreases seen in most other orthologs. Reversion of the atypical tryptophan in the S4 sequence (RWIR→RRIR) inverted the optical polarity, producing a small but clear decrease in fluorescence. This divergence underscores that even a single side-chain substitution can alter how S4 motion can influence the response of the fused fluorescent protein. The contrasting responses of hamster wildtype and W→R mutant therefore illustrate the fine resolution of GEVI readouts: they not only detect whether S4 moves but also reveal differences in the orientation and/or trajectory of the FP domain during conformational changes. Such insights highlight the unique ability of GEVIs to probe the mechanics of voltage-dependent protein motion beyond what electrical recordings alone can resolve.

Among the responsive constructs, three stood out. The plankton VSD exhibited a left-shifted voltage dependence, producing clear responses at relatively modest depolarizations (Figure 2). This suggests that environmental pressures in plankton may have selected for different voltage ranges and highlights the potential of nonvertebrate orthologs other than *Ciona* as sources of novel GEVI properties. By contrast, the human VSD (hVSP2) produced weak, sluggish signals that failed to recover after repolarization. This finding is notable because a previous report demonstrated membrane expression for hVSP2 but could not detect sensing charges [10]. Whether the presence of the phosphatase domain prevents S4 movement, or whether the unusual motion of the human VSD’s S4 helix does not involve charges traversing the voltage field, remains unclear. Reverting a primate-specific S1 mutation (G95R) partially restored signal size and repolarization recovery, while also accelerating onset, suggesting that the glycine substitution destabilized conformational reversibility (Figure 4). Because this residue lies near the membrane-cytosol interface, these results raise the possibility that primate VSPs have adapted altered interactions with membrane lipids, with consequences for VSD conformational dynamics.

The third standout was the reduced but still significant depolarization required for the sea hare construct. This is notable because of the amino acid at position 217 in S4. In *Ciona* phosphatase and several GEVI constructs, the R217Q mutation shifts the voltage response toward negative potentials by over 100 mV, enabling detection of hyperpolarization steps [7,11,12]. Although sea hare also retains four sensing charges in S4, like *Ciona*, its requirement for modest depolarization suggests that additional VSD interactions can modify the voltage-shifting influence of 217Q.

Together, these findings underscore the modularity and tunability of VSP VSDs [9]. Chimeric swaps showed that the N-terminal region has little influence on voltage sensing itself, whereas the S4 composition and S1 lipid-facing residues [20] strongly impact kinetics and recovery (Figure 3, Figure 4 and Figure 5). Natural variants thus provide a complementary approach to mutagenesis for uncovering biophysical principles of voltage sensing. For GEVI design, this comparative strategy offers a way to identify and harness favorable combinations of residues, linkers, and domains to achieve desired response properties.

Although this work represents an initial survey, the cross-species comparison establishes a quantitative foundation for modeling how specific residues and electrostatic networks tune voltage dependence across the VSP family. The results underscore that even within a conserved scaffold, sequence diversity can yield distinct voltage ranges and kinetics, providing a resource for computational analyses and targeted mutagenesis.

In addition to the single-molecule behavior emphasized here, previous studies have shown that VSPs can form functional dimers [17], suggesting that interdomain interactions could influence optical polarity and trafficking efficiency. Differences in dimerization propensity or in N-terminal trafficking motifs may therefore contribute to the diverse responses observed among orthologs. These possibilities provide an intriguing direction for future work combining voltage-clamp fluorometry with biochemical assays of oligomerization.

More broadly, by linking evolutionary sequence variation to measurable changes in voltage sensitivity, this study defines a framework for systematic exploration of structure–function relationships within the VSP family. Such quantitative comparisons will guide both mechanistic studies of native VSP biology and the rational design of next-generation GEVIs.

Two non-exclusive models emerge from our cross-species survey. First, an attenuated voltage constraint model: if membrane voltage no longer exerts strong selective pressure on VSPs, the VSD may drift, broadening operating ranges and kinetics without abolishing responsiveness. Second, a persistent voltage constraint model: despite substantial sequence divergence, every construct that reached the plasma membrane produced a voltage-dependent optical signal with a measurable V_1/2_, arguing that voltage sensing remains functionally relevant to VSP biology. Our data support elements of both views: the spread of V_1/2_ values and kinetics is consistent with relaxed tuning, whereas the retention of voltage coupling—including in unusual variants such as hamster RWIR and primate S1—argues against complete neutral drift of the VSD.

Mechanistically, several forces could reconcile these models. (i) Electrostatic network degeneracy: Multiple residue constellations across S1–S4 can yield a voltage-coupled conformational change, allowing substantial sequence diversity while preserving function. This principle is exemplified by the hamster variant, where R-W-I-R behaves distinctly from the canonical R-R-I-R (Figure 5). (ii) Contextual coupling: Differences in lipid interaction, dimerization propensity, and linker geometry can re-map similar VSD motions onto different optical outputs, preserving voltage dependence but shifting apparent V_1/2_ and kinetics. The human S1 mutation provides a potential example of altered lipid interaction (Figure 4). (iii) Division of labor across domains: If the phosphatase active site experiences lineage-specific constraints (substrate profile, localization, partner proteins), voltage sensing may be retained as a gating input but re-tuned to each cellular niche which may account for broad voltage range of different species shown in Figure 2. These evolutionary models highlight the enduring role of voltage in VSP biology, but they also underscore the need to examine how VSD motion couples to the native phosphatase domain—an aspect necessarily excluded by our GEVI design.

A caveat of this study is that the native phosphatase domain was replaced with a fluorescent protein (Figure 1). While this design allowed us to directly monitor VSD motion through optical signals, it necessarily removed the ability to assess how voltage sensing is coupled to phosphatase activity. As a result, the conclusions here apply specifically to VSD conformational dynamics and their optical readouts, rather than to the full enzymatic cycle of VSPs. Indeed, the linker region in these constructs has been shown to interact with the plasma membrane [21] and that the linker region can also interact with the phosphatase domain [22]. It is unclear the effect of replacing the phosphatase domain with a fluorescent protein has on these observed properties of the linker section.

This limitation can be addressed with complementary strategies that preserve enzymatic coupling. One approach is voltage-clamp fluorometry of full-length VSPs, in which an environmentally sensitive dye attached to a cysteine near S4 reports conformational changes while the native phosphatase domain remains intact [12,20]. Another is to use VSD-only ASAP scaffolds, where cpGFP is inserted into the S3–S4 loop to monitor VSD motion independent of the enzyme [23].

In conclusion, our cross-species analysis demonstrates that the capacity for voltage-dependent optical signals is deeply conserved across the VSP family, though its expression varies depending on host compatibility and evolutionary sequence differences. These results establish GEVIs as a powerful model not only for dissecting voltage sensing but also for revealing potential cellular influences on protein activity. For instance, the addition of a trafficking motif altered GEVI activity, suggesting that interaction of the trafficking partner may persist at the plasma membrane [24]. GEVI utilization may now expand into probing protein–lipid interactions, providing new opportunities for engineering GEVIs with tailored kinetics and voltage ranges.

## 4. Materials and Methods

### 4.1. In Silico Search Strategy

The amino acid sequence from the voltage-sensing domain (S1–S4) of the *Monsiga brevicollis* VSP (XP_001743274.1) was used as a bait sequence in a PHI BLAST (v2.17.0) search (National Center for Biotechnology Information (NCBI). Basic Local Alignment Search Tool (BLAST). Available from: https://blast.ncbi.nlm.nih.gov/, accessed on 23 September 2025) requiring the presence of the following amino acid pattern: [FYW]xx[E,D]xxx[R,K], where x is any amino acid [25]. Alignments were performed by Clustal Omega v1.2.4 [26] using the percent identity matrix to remove redundant sequences. Logo consensus plots were generated by the Weblogo v3.9.0 (https://weblogo.berkeley.edu, accessed on 23 September 2025) [15].

### 4.2. Plasmid Design and Construction

Synthesized DNA (Integrated DNA Technologies, Coralville, IA, USA) was cloned into a pcDNA 3.1 (Invitrogen, Waltham, MA, USA) containing the fluorescent protein Super Ecliptic pHlourin A227D. For chimera generation, primers were designed to introduce point mutations to the S0 domain as required. Conventional one-step and two-step PCR were used to generate the target inserts. The inserts were cloned into the vector using NEB restriction enzymes. Two-step PCR was also used to generate point mutations as well as chimeric constructs.

### 4.3. Cell Culture and Transfection

HEK 293 cells were obtained from the American Type Culture Collection (ATCC, Manassas, VA, USA) and cultured in Dulbecco’s Modified Eagle Medium (DMEM; Gibco, Waltham, MA, USA) supplemented with 10% Fetal Bovine Serum (FBS; Gibco). For transfection, HEK 293 cells were suspended using 0.25% Trypsin-EDTA (Gibco) then plated onto poly-L-lysine (Sigma-Aldrich, St. Louis, MO, USA) coated #0 coverslips (Ted Pella, Redding, CA, USA). Transient transfection was carried out with Lipofectamine 2000 (Invitrogen) according to the manufacturer’s protocol.

### 4.4. Electrophysiology

Coverslips with transiently transfected cells were placed into a patch chamber (Warner instruments, Holliston, MA, USA) sealed with a #0 thickness cover glass for simultaneous voltage clamp and fluorescence imaging. The chamber was kept at 34 °C throughout the experiment and perfused with extracellular solution (150 mM NaCl, 4 mM KCl, 1 mM MgCl_2_, 2 mM CaCl_2_, 5 mM D-glucose and 5 mM HEPES, pH = 7.4). Filamented glass capillary tubes (1.5 mm/0.84 mm; World Precision Instruments, Sarasota, FL, USA) were pulled by a micropipette puller prior to each experiment to resistances of 3–5 MΩ for HEK 293 cells. The pipettes were filled with intracellular solution (120 mM K-aspartate, 4 mM NaCl, 4 mM MgCl_2_, 1 mM CaCl_2_, 10 mM EGTA, 3 mM Na_2_ATP and 5 mM HEPES, pH = 7.2) and held by a pipette holder (HEKA) mounted on a micromanipulator (Scientifica, Uckfield, UK). Whole cell voltage clamp of transfected cells were conducted using a patch clamp amplifier (HEKA, Lambrecht, Germany).

### 4.5. Fluorescence Microscopy of Cultured Cells

An inverted microscope (IX71; Olympus, Tokyo, Japan) equipped with a 60× oil-immersion lens, 1.35-numerical aperture (NA), was used for epifluorescence imaging. The light source was a 75 W Xenon arc lamp (Osram, Munich, Germany) placed in a lamp housing (Cairn, Edinburgh, UK). GFP was imaged using a filter cube consisting of an excitation filter (FF02-472/30-25), a dichroic mirror (FF495-Di03) and an emission filter (FF01-496) for the 470 nm wavelength excitation all from Semrock (New York, NY, USA). Two cameras were mounted on the microscope through a dual port camera adapter (Olympus). A color CCD camera (Hitachi, Chiyoda, Japan) was used to visualize cells during patch clamp experiments. Fluorescence of the voltage indicators were recorded at 1 kHz frame rate by a high-speed CCD camera (RedShirtImaging, Decatur, GA, USA). All optical devices were placed on a vibration isolation platform (Kinetic systems, Boston, MA, USA) to avoid any vibrational noise during patch clamp fluorometry experiments.

### 4.6. Confocal Microscopy

Before imaging, the DMEM present in the confocal dishes was replaced with 200 µL of 1× Phosphate-Buffered Saline (Tech & Innovation, Chuncheon, Republic of Korea). Confocal images were then acquired using an A1R laser scanning confocal microscope (Nikon, Tokyo, Japan) and a 60× objective oil lens (Plan Apo λ 60×; Nikon). A 488 nm laser (Nikon, Tokyo, Japan) was used for excitation and a 525/50 nm dichroic mirror was used for detection.

### 4.7. Molecular Structure Models

Crystal structures or Alphafold predictions were visualized using the UCSF Chimera v1.15 program [27]. Overlays of structures were obtained using the Matchmaker function.

## Figures and Tables

**Figure 1 ijms-26-10963-f001:**
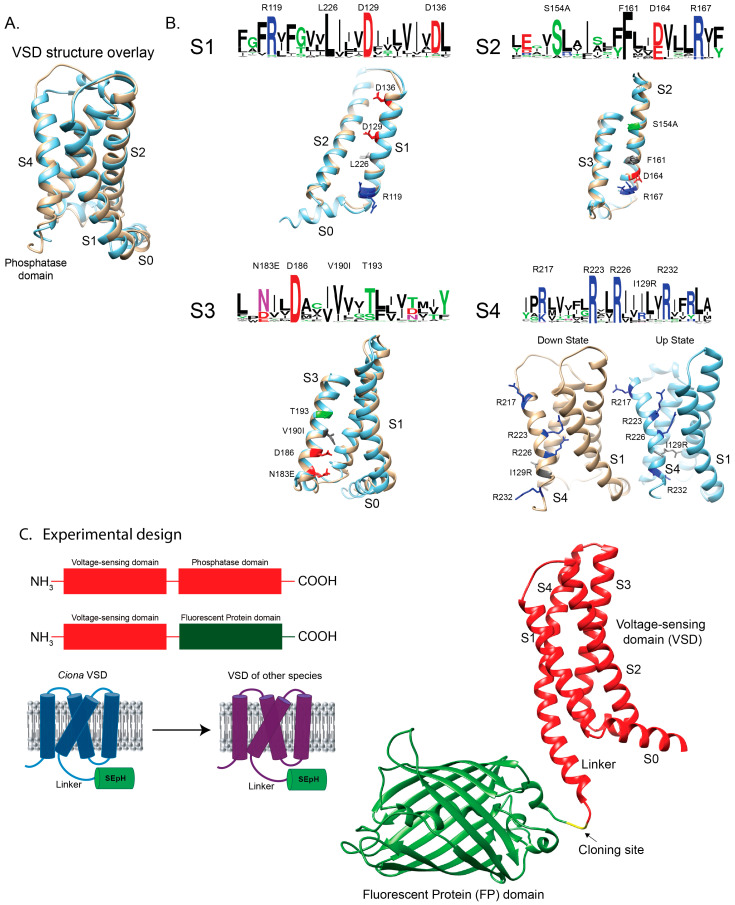
Exploring sequence variation in voltage-sensing proteins. (**A**) Structural overlay of the down state (brown, PDB 4g7v) and up state (blue, PDB 4g80) of the *Ciona intestinalis* VSD [13]. Trans-membrane segments are labeled S1, S2, and S4 (S3 not labeled for clarity). S0 is a cytosolic α-helix immediately preceding S1. (**B**) Amino acid conservation logos for S1–S4. The relative size of each residue indicates conservation; red denotes negatively charged, blue positively charged, green and purple polar, and black nonpolar amino acids. Highly conserved residues are mapped onto the structures in matching colors. Where *Ciona* differs from the consensus, labels indicate the consensus residue, its position, and the amino acid present in *Ciona* (e.g., S154A). (**C**) Experimental design schematic. The phosphatase domain of each VSP ortholog was replaced with a fluorescent protein, leaving the VSD intact to monitor plasma membrane localization and voltage-dependent fluorescence changes. At right, an AlphaFold3 [14] model shows a representative construct, with the N-terminal random coil omitted for clarity (cloning site in yellow).

**Figure 2 ijms-26-10963-f002:**
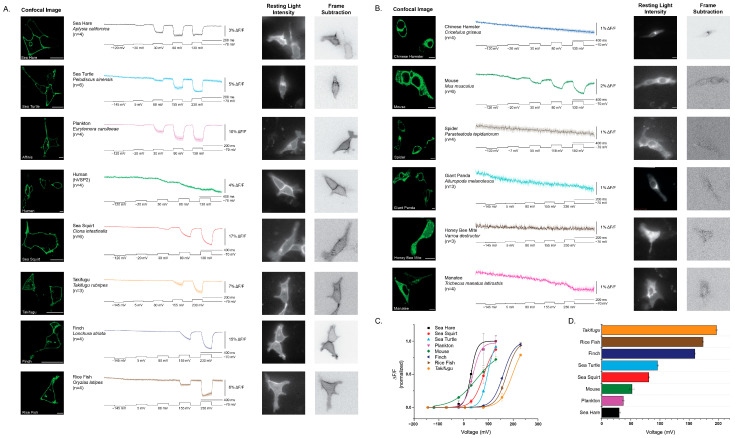
Expression and voltage-dependent responses of VSP–FP chimeras across species. (**A**) Representative fluorescence traces from HEK 293 cells expressing VSP constructs that trafficked efficiently to the plasma membrane. The central traces show the mean ΔF/F responses (solid line) with standard error (shaded area) during voltage steps (black trace below). Voltage protocols were adjusted for each species to maximize signal amplitude. Resting light images show widefield fluorescence from representative cells under whole-cell voltage clamp. Frame-subtraction images depict the difference between fluorescence during the largest depolarization step and baseline, highlighting the spatial origin of voltage-dependent signals and thus guiding region of interest determination. (**B**) Constructs with limited plasma-membrane localization and high intracellular accumulation. Although most failed to generate clear responses, the mouse construct produced a detectable voltage-dependent signal despite reduced surface expression. (**C**) Normalized fluorescence–voltage (ΔF/F vs. V) relationships. Individual traces were fitted with a Boltzmann function to estimate maximal signal size, and the resulting fits were used to normalize relative fluorescence changes across species. (**D**) Comparison of half-activation voltages (V_1⁄2_) derived from the Boltzmann fits in (**C**). Constructs that reached the plasma membrane exhibited a wide range of voltage sensitivities, from left-shifted (plankton and sea hare) to highly depolarized (finch and turtle). Scale bars are 10 μm.

**Figure 3 ijms-26-10963-f003:**
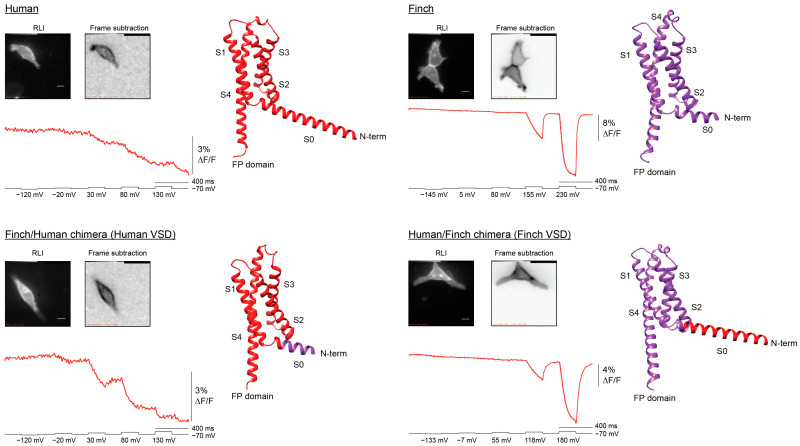
The voltage-sensing domain determines species-specific optical responses. Representative data from Human VSD (hVSP2), Finch VSD, and reciprocal Human/Finch chimeras. Both chimeras were generated with the N-terminus fused in the S0 helix. Resting light images (RLI) show robust plasma membrane localization across constructs (scale bars, 10 µm). Frame subtraction images highlight regions of stimulus-dependent optical change, calculated as the difference between baseline (50 frames after hyperpolarization) and the response to the strongest depolarization step (50 frames). Fluorescence traces are shown for each construct during voltage steps (black trace, −70 to +130 mV for Human, extended to stronger depolarizations for Finch). The Human VSD produces a sluggish depolarization response with poor recovery, whereas the Finch VSD produces a rapid biphasic response but requires much stronger depolarization to activate. Importantly, both chimeras containing the Human VSD resembled each other, while both chimeras containing the Finch VSD resembled each other, demonstrating that the VSD sequence is responsible for the polarity and kinetics of the optical signal. AlphaFold-predicted structures [14] of the chimeras are shown, with Human sequence segments in red and Finch segments in purple. For clarity, the models omit the N-terminus and the fused fluorescent protein (FP) domain.

**Figure 4 ijms-26-10963-f004:**
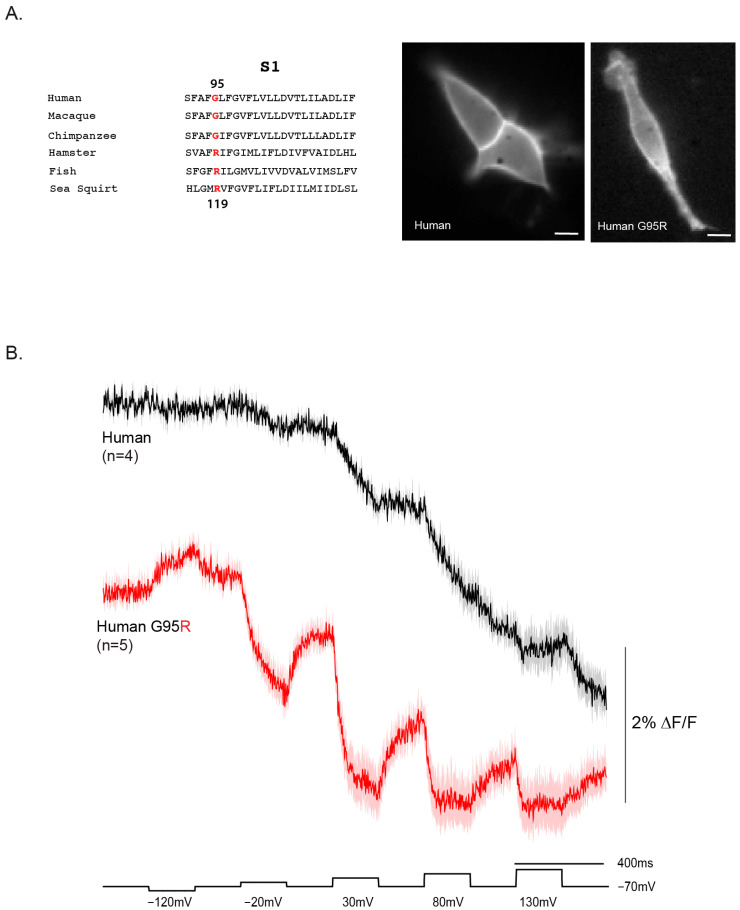
A primate-specific mutation in human VSP alters voltage-dependent responses. (**A**) Sequence alignment of VSP orthologs highlights position 95 in S1 (red), with top numbering based on the human sequence and bottom numbering based on the *Ciona* sequence. Most species retain an arginine at this position, whereas humans and other primates have glycine. Resting light fluorescence images of wildtype human VSD–FP and the G95R mutant show robust plasma membrane localization with minimal intracellular accumulation (scale bars, 10 µm). (**B**) Representative fluorescence traces recorded during depolarizing voltage steps (−120 mV to +130 mV; black trace shown below). The wildtype human construct produces a sluggish depolarization response with poor recovery, whereas the G95R mutant shows partial restoration of the repolarization phase along with improved signal size and faster kinetics.

**Figure 5 ijms-26-10963-f005:**
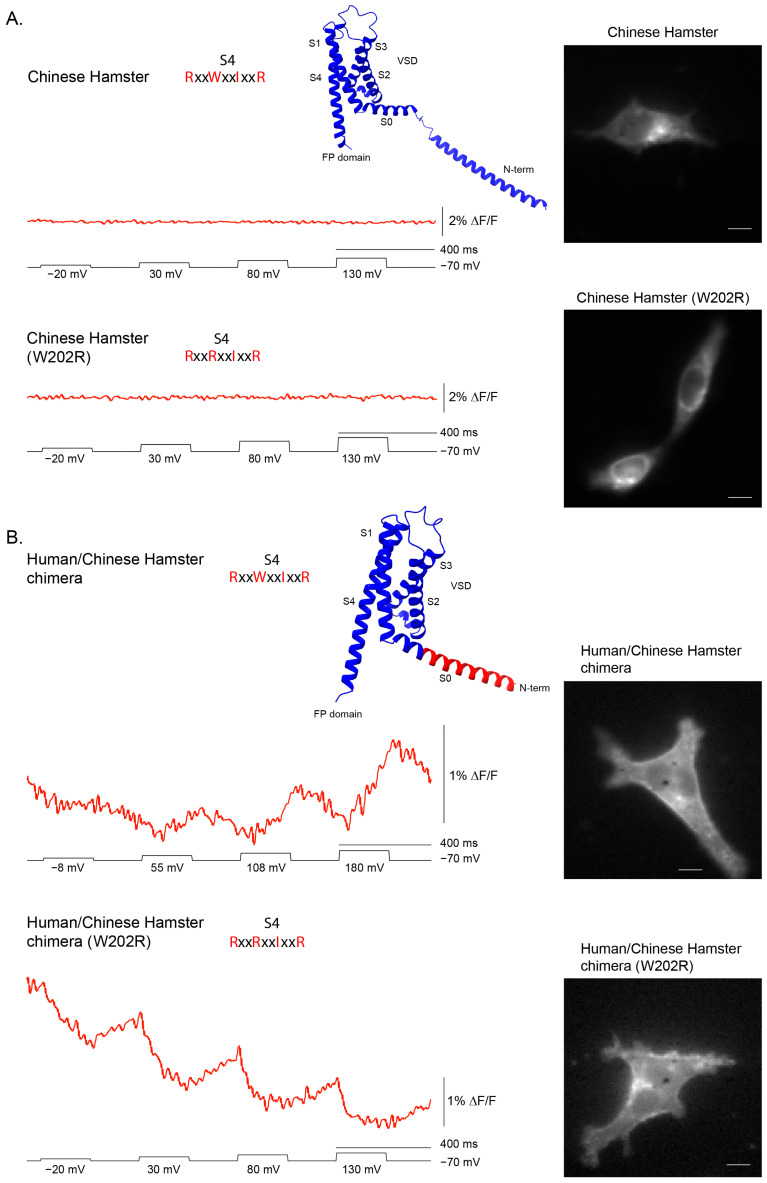
Human–hamster chimeras reveal how S4 composition shapes the optical signal. (**A**) Representative fluorescence traces are shown for wildtype hamster VSD–FP and the hamster W202R mutant (numbering based on the hamster sequence). (**B**) Representative fluorescence traces are shown for the corresponding human/hamster chimeras. The wildtype hamster construct produces little or no signal, whereas the human/hamster chimera with the native hamster S4 sequence (RWIR) shows a small increase in fluorescence upon depolarization. Introducing the W202R substitution in this chimera inverts the polarity of the voltage-dependent signal, indicating that the second position of S4 critically influences how conformational changes alter to the fluorescent protein response. Structural models predicted by AlphaFold are shown, with hamster sequence in blue and human sequence in red. Scale bar is 10 μm.

**Table 1 ijms-26-10963-t001:** Voltage-dependent optical properties of GEVIs with VSDs from different species. For each construct, the largest fluorescence change (ΔF/F) and the corresponding depolarization step are listed. The on (τ on) and off (τ off) kinetics are reported for the largest depolarization step, determined by fitting the optical signal to a single exponential decay. Where both fast and slow components are shown, the response was best fit by a double exponential. Percentages depict the relative amplitudes of each component. (N.D.—not determined).

Species	Tau On (ms)	Tau Off (ms)	V_1/2_ (mV)	Largest ΔF/F
Sea Hare	12.8 ± 1.2	9.6 ± 0.2	26.3 ± 2.9	3%/150 mV
Sea Turtle	30.1 ± 0.3	11.5 ± 1.4	85.2 ± 0.4	4.5%/300 mV
Rice Fish	7.9 ± 0.2	3.8 ± 0.1	>150	5.8%/300 mV
Plankton	8.7 ± 1.1	13 ± 0.3	37.9 ± 1.5	9.7%/200 mV
Finch	49.3 ± 1.7	9.1 ± 1.0 (fast—68.5%)	>160	15%/300 mV
		35.4 ± 1.1 (slow—31.5%)		
Sea Squirt	20.7 ± 0.7 (fast—51%)	15.9 ± 0.4 (fast—87%)	80.7 ± 1.7	15.5%/200 mV
	103.3 ± 8.5 (slow—49%)	133.1 ± 31 (slow—13%)		
Mouse	67.6 ± 3.1	37.3 ± 1.5	56.2 ± 3.6	1.8%/200 mV
*Takafugu*	30.8 ± 1.9	6.75 ± 0.1	>190	6.5%/300 mV
Human	131.6 ± 3.0	N.D.	N.D.	2.5%/150 mV

## Data Availability

The raw data supporting the conclusions of this article will be made available by the authors on request.

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
