# Peer review of "From Plankton to Primates: How VSP Sequence Diversity Shapes Voltage Sensing"

_ijms, 2025, doi:10.3390/ijms262210963_

Round 1

Reviewer 1 Report

Comments and Suggestions for Authors

The current manuscript examines the voltage-sensing properties of a large diversity of voltage-sensitive phosphatase (VSP) across the animal kingdom.

This is exciting new work that can provide novel insights into the molecular mechanisms determining voltage sensing in VSPs (and possibly beyond) and provide initial insights into the poorly understood biology of these unique enzymes. The approach is straightforward and technically sound.

Interesting observations include striking diversity in voltage sensitivity and kinetics as well as the initial identification of some molecular determinants that affect the differential behavior.

This said, the study appears somewhat preliminary, but could easily gain a lot more substance by a more quantitative analysis.

Specific points:

  1. The voltage dependency is discussed as one of the major distinguishing features among the VSPs, however, it is not analyzed properly. Provide quantitative measures of the voltage sensitivity (fluorescence-voltage curves, V1/2…). It should be straight-forward to derive these parameters from the data traces shown.

  1. 2A: labeling of the voltage protocols may be wrong: e.g., for sea hare: really progressing from step to -80 mV to +130 mV? The direction as well as size of the fluorescent signals also do not make sense (hyperpolarizing from holding -70 to -80 should not produce the same decrease as depolarizing in the subsequent steps). At the very least, the voltage trace (even if only meant to be schematic) is inconsistent with the indicated potentials. Same for some of the other voltage protocols. Overall, the distinct voltage protocols indicated for the different VSP orthologs are pretty confusing and certainly preclude meaningful comparison of the voltage sensitivities based on the fluorescence traces (cf. point 1).

  1. Kinetics: the finding that VSPs strongly differ in their activation and deactivation kinetics is new, and is even addressed mechanistically (Fig.4). Why hide the quantification in a table in the supplement? A graphic display (at least for some characteristic orthologs) should be provided in the main paper.

  1. The slow kinetics seen in some lineages is interesting, specifically for the human VSP. While it is noted that ‘the fluorescence signal does not return to baseline’, it seems that the kinetics are too slow to be resolved with the voltage protocol used. Obviously at this point an altered voltage step protocol with longer step duration is needed, both for the ON an the OFF response, rather than sticking to the same protocol that works well with the faster VSP orthologs.

  1. Linker sequence: the design of the constructs appears to include the ‘linker’ just N-terminal to the fluorescent reporter (should be explicitly included in the scheme Fig. 1C). Importantly, the linker is shown to impact substantially on electrochemical coupling (recent study: Mizutani et al., PNAS 2025) and may thus affect voltage sensitivity. Therefore the linker sequence should be considered, when comparing VSPs with distinct functional properties.

  1. The presentation requires a much more informed approach/presentation of the phylogenetic aspect, since this is the strength of the study. Thus, phylogenetic relations need to be clearly outlined (e.g. looking at common properties of mammalian, or vertebrate VSPs, etc.). Likewise the logic behind the choice of the species needs to be made more clear, as multiple samples are very close relatives (e.g. various mammalian VSPs), whereas others are very distant (arthropods, mollusk, unicellular organism). Moreover, the source species need to be given properly. Most disconcerting, ‘plankton’ is certainly not a name that would allow to even approximately identify the phylogenetic branch of the animal kingdom (if it is an animal at all).
  2. The ms cannot decide whether it looks for useful properties for engineering GEVIs or for the behavior of (native) VSPs. A clear perspective would help understanding the experimental approach (and still allow to discuss both aspects in the Discussion section). In this reviewer’s view, the biology/mechanism is much more interesting than the use for GEVI design.

Minor points

  1. Abstract, l.5: ‘optical signal’: the approach for recording voltage responses is not described in the abstract, so this is actually difficult to understand when only reading the abstract.
  2. Abstract, l.5: ‘across the family’, sightly misleading as there seems to be only one ortholog in each genome (except for primates), so there is not really a family of VSPs.
  3. human VSP: not clear if this is human VSP1 or VSP2. This should clearly stated throughout the entire manuscript.
  4. Abstract , .l14: ‘coupling between VSD motion and the fused fluorescent protein’: Is ‘coupling’ what is analyzed ere? It seems in most of the manuscript the assumption is that the assay reports voltage sensitivity of the VSD.
  5. ‘membrane expression’: although this term has been used widely, it is actually incorrect; I suggest to replace by membrane localization, targeting, or abundance.
  6. What is the yellow sequence stretch in Fig. 1C? I this an additional (non-VSP/non FP) sequence introduced?
  7. 5, l.3: zebrafish, reported in the text is actually not included in the data (Fig. 2).
  8. 2B: fluorescence scale bar is missing for the first few examples (required even if there is no voltage-dependent signal)
  9. Voltages applied are quite extreme for many experiments. Fluorescence changes may also result from damage to the cells. Please provide some indication of what input resistances were considered acceptable for including data.
  10. p 8. ‘If the S0 helix influences voltage-dependent signaling, its effect is minimal.’. The conclusion appears imprecise. The experiments indicate that the differences in S0 between the orthologs examined do not explain the different kinetic behavior; however, S0 could still be important, in particular its membrane-proximal part that was not exchanged.
  11. 10 ‘kinetics improved’: accelerated?
  12. 10/Fig.5 : Reversal of fluorescence signals is really interesting. Does this indicate reversal of VSD movement. More in-depth analysis would be valued.
  13. Discussion: suggestion that distinct membrane targeting may depend on hist cell: while this appears plausible, there are no data to support this conclusion.
  14. 13, l.1: ‘the unusually strong depolarization required for the sea hare construct.’ According to the data shown in Fig.1A, this is not the case (e.g., compare with Takifugu trace)
  15. Materials and Methods: 4.1 ‚ Monsigia brevicollis‘ should read ‘Monosiga brevicollis’? This is a choanoflagellate. Context is missing.
  16. Materials and Methods: 4.5: Description of analysis of the fluorescence signal is essentially missing (choice of ROIs, background subtraction, etc…).

Author Response

Dear Editor,

We sincerely thank the reviewers for their insightful feedback. The revised manuscript now includes: (i) quantitative ΔF/F–V analysis with V₁⁄₂ values (Figures 2C–D, Table 1); (ii) corrected voltage protocols; (iii) updated Figures 1–2 with linker labels and scientific species names; (iv) a clarified Introduction specifying the study aim; and (v) an expanded Discussion addressing evolutionary models, dimerization, and contextual coupling. These revisions substantially improve the manuscript’s rigor and clarity. Below are the reviewers’ comments with our replies in red.

The current manuscript examines the voltage-sensing properties of a large diversity of voltage-sensitive phosphatase (VSP) across the animal kingdom.

This is exciting new work that can provide novel insights into the molecular mechanisms determining voltage sensing in VSPs (and possibly beyond) and provide initial insights into the poorly understood biology of these unique enzymes. The approach is straightforward and technically sound.

We thank the reviewer for this encouraging assessment. We agree that a more quantitative presentation strengthens the manuscript. Accordingly, we now provide Boltzmann fits of ΔF/F versus voltage for multiple orthologs and report the corresponding V₁₂ values (Figure 2C–D, Table 1). These additions enable direct comparison of voltage sensitivities across species.

Interesting observations include striking diversity in voltage sensitivity and kinetics as well as the initial identification of some molecular determinants that affect the differential behavior.

This said, the study appears somewhat preliminary, but could easily gain a lot more substance by a more quantitative analysis.

Specific points:

  1. The voltage dependency is discussed as one of the major distinguishing features among the VSPs, however, it is not analyzed properly. Provide quantitative measures of the voltage sensitivity (fluorescence-voltage curves, V1/2…). It should be straight-forward to derive these parameters from the data traces shown.

Addressed as described above. We generated normalized ΔF/F–V plots and fitted them with Boltzmann functions to estimate V₁₂ for each construct. These analyses are presented in Figures 2C–D and Table 1.

  1. 2A: labeling of the voltage protocols may be wrong: e.g., for sea hare: really progressing from step to -80 mV to +130 mV? The direction as well as size of the fluorescent signals also do not make sense (hyperpolarizing from holding -70 to -80 should not produce the same decrease as depolarizing in the subsequent steps). At the very least, the voltage trace (even if only meant to be schematic) is inconsistent with the indicated potentials. Same for some of the other voltage protocols. Overall, the distinct voltage protocols indicated for the different VSP orthologs are pretty confusing and certainly preclude meaningful comparison of the voltage sensitivities based on the fluorescence traces (cf. point 1).

We are grateful for catching this. All voltage protocols were verified and corrected. Because individual orthologs required different voltage ranges to elicit signals, the revised figure shows distinct step protocols for clarity. The normalized ΔF/F–V plots now allow meaningful cross-species comparison.

Note on data inclusion:
The horseshoe crab and tick constructs were removed from the revised manuscript because their fluorescence responses were non-uniform across the plasma membrane and could not be analyzed consistently alongside the other orthologs. These behaviors may reflect distinct trafficking or assembly properties that warrant separate investigation. We therefore plan to present these results in a future study focused specifically on spatially heterogeneous membrane responses.

  1. Kinetics: the finding that VSPs strongly differ in their activation and deactivation kinetics is new, and is even addressed mechanistically (Fig.4). Why hide the quantification in a table in the supplement? A graphic display (at least for some characteristic orthologs) should be provided in the main paper.

We have moved the kinetic data into the main text and combined them with the new V₁/₂ values (Table 1).

  1. The slow kinetics seen in some lineages is interesting, specifically for the human VSP. While it is noted that ‘the fluorescence signal does not return to baseline’, it seems that the kinetics are too slow to be resolved with the voltage protocol used. Obviously at this point an altered voltage step protocol with longer step duration is needed, both for the ON an the OFF response, rather than sticking to the same protocol that works well with the faster VSP orthologs.

We extended the voltage step duration for hVSP2 by twofold but still did not reach a steady plateau. The revised text explains that hVSP2 kinetics remain unusually slow, and we now emphasize that our goal is to demonstrate the presence of a voltage-dependent signal rather than to extract precise kinetic constants.

  1. Linker sequence: the design of the constructs appears to include the ‘linker’ just N-terminal to the fluorescent reporter (should be explicitly included in the scheme Fig. 1C). Importantly, the linker is shown to impact substantially on electrochemical coupling (recent study: Mizutani et al., PNAS 2025) and may thus affect voltage sensitivity. Therefore the linker sequence should be considered, when comparing VSPs with distinct functional properties.

This is an excellent point and the reason we suggested using ASAP GEVIs to monitor potential effects of the phosphatase domain. In the Mizutani et al., PNAS 2022 they show that S4 can interact with the phosphatase domain. They also show that the linker can interact with the plasma membrane which was also shown by Kohout et al. (2010). This degree of complexity raises very interesting questions that we would like to address in the future. Indeed, our GEVI, Bongwoori-R3, resulted from mutagenesis to the linker region. Figure 1C now includes labeled linker regions as requested. We have also added these words to the discussion: Indeed, the linker region in these constructs has been shown to interact with the plasma membrane [20] and that the linker region can also interact with the phosphatase domain [21]. It is unclear the effect of replacing the phosphatase domain with a fluorescent protein has on these observed properties of the linker section.

Including a reference to the Mizutani paper.

  1. The presentation requires a much more informed approach/presentation of the phylogenetic aspect, since this is the strength of the study. Thus, phylogenetic relations need to be clearly outlined (e.g. looking at common properties of mammalian, or vertebrate VSPs, etc.). Likewise the logic behind the choice of the species needs to be made more clear, as multiple samples are very close relatives (e.g. various mammalian VSPs), whereas others are very distant (arthropods, mollusk, unicellular organism). Moreover, the source species need to be given properly. Most disconcerting, ‘plankton’ is certainly not a name that would allow to even approximately identify the phylogenetic branch of the animal kingdom (if it is an animal at all).

Clarified. We now explain our selection rationale: species were chosen to test whether voltage remains an evolutionary constraint on VSPs primarily based on sequence divergence. All organism names are now given in full (e.g., Eurytemora carolleeae, Aplysia californica, Cricetulus griseus) in the supplemental and in figure 2A. “Plankton” has been replaced by the proper taxonomic identification. The revised Introduction and Methods 4.1 describe this framework.

  1. The ms cannot decide whether it looks for useful properties for engineering GEVIs or for the behavior of (native) VSPs. A clear perspective would help understanding the experimental approach (and still allow to discuss both aspects in the Discussion section). In this reviewer’s view, the biology/mechanism is much more interesting than the use for GEVI design.

This is a very fair point. On the GEVI design side, we do a few mutations and ask how did that change the voltage. This approach was motivated by asking what has Nature done and how has that affected the voltage response. This small sampling was truly fascinating and revealed to us that GEVIs can be useful for more than simply monitoring voltage transients. We have now clarified in the Introduction that this work is motivated by both perspectives—understanding VSP biology and informing GEVI engineering—and that these aims are complementary rather than mutually exclusive.

Minor points

  1. Abstract, l.5: ‘optical signal’: the approach for recording voltage responses is not described in the abstract, so this is actually difficult to understand when only reading the abstract.

We’ve added ‘by replacing the phosphatase domain with a fluorescent protein to enable optical detection of VSD responses’ and ‘during modest depolarizations of the plasma membrane.’ to the abstract for clarification.

  1. Abstract, l.5: ‘across the family’, sightly misleading as there seems to be only one ortholog in each genome (except for primates), so there is not really a family of VSPs.

We changed family of VSPs to VSP othologs in the abstract

  1. human VSP: not clear if this is human VSP1 or VSP2. This should clearly stated throughout the entire manuscript.

We have added (hVSP2) throughout the manuscript and to figure 2.

  1. Abstract , .l14: ‘coupling between VSD motion and the fused fluorescent protein’: Is ‘coupling’ what is analyzed ere? It seems in most of the manuscript the assumption is that the assay reports voltage sensitivity of the VSD.

A good point. The fluorescence change is voltage-dependent indicating that movement of the VSD mediates the fluorescence transition. However, we do not think the coupling has been altered. Rather the fluorescence response is altered because the VSD has altered the positioning/movement of the FP domain. We therefore reworded the abstract: ‘These contrasting behaviors show that single residue changes can reshape how VSD movements influence the fluorescent reporter, highlighting the molecular precision revealed by GEVI measurements.’

  1. ‘membrane expression’: although this term has been used widely, it is actually incorrect; I suggest to replace by membrane localization, targeting, or abundance.

A good point. Corrected.

  1. What is the yellow sequence stretch in Fig. 1C? I this an additional (non-VSP/non FP) sequence introduced?

Thank you for noticing that. That is the restriction site used to clone the FP domain downstream of the VSD. Words describing this have been added to the legend.

  1. 5, l.3: zebrafish, reported in the text is actually not included in the data (Fig. 2).

We are very grateful for noticing this. The zebrafish GEVIs have already been published. Removed.

  1. 2B: fluorescence scale bar is missing for the first few examples (required even if there is no voltage-dependent signal)

Corrected

  1. Voltages applied are quite extreme for many experiments. Fluorescence changes may also result from damage to the cells. Please provide some indication of what input resistances were considered acceptable for including data.

Yes, a good point. Each cell was subjected to 4 trials of the voltage step protocol with each trial consisting of 4 repetitions of the voltage step protocol. This enabled us to monitor the integrity of the seal over the four trials. Cells that showed consistent current injections and fluorescent reponses for each trial were considered for analysis.

  1. p 8. ‘If the S0 helix influences voltage-dependent signaling, its effect is minimal.’. The conclusion appears imprecise. The experiments indicate that the differences in S0 between the orthologs examined do not explain the different kinetic behavior; however, S0 could still be important, in particular its membrane-proximal part that was not exchanged.

A very good point. We have changed the wording to say that any effect of S0 was not noticeable via this method.

  1. 10 ‘kinetics improved’: accelerated?

Not necessarily. The optical signal involves interaction between neighboring FP domains. An increase in the optical response speed could mean that the FP domains are in a different orientation that mediates a faster optical response. 

  1. 10/Fig.5 : Reversal of fluorescence signals is really interesting. Does this indicate reversal of VSD movement. More in-depth analysis would be valued.

This is a great question. We have a publication that just came online (Leong, Shin et al., 2025 ACS Sensors) that explores the different optical signals given by the fluorescent protein depending on the interaction between neighboring FP domains. In that publication, we demonstrate that different FP interactions can alter the fluorescent output depending on the flexibility of the chromophore. That paper also shows quite nicely that membrane curvature can sometimes have an effect. So if the movement of the FP is altered, it can cause different optical signals. Therefore, the short answer to the reviewer’s question is no, we do not think that the movement is reversed (though we cannot rule out that possibility) instead, we think the movement and/or the resting state interaction between FP domains is altered.

  1. Discussion: suggestion that distinct membrane targeting may depend on hist cell: while this appears plausible, there are no data to support this conclusion.

We agree that our data do not directly test the influence of host cell type on membrane targeting. The statement has been revised to clarify that cell-specific factors are only a possible, not demonstrated, contributor. The revised text now reads:

“Our results show that while the capacity to generate voltage-dependent optical signals is broadly conserved, differences in expression, kinetics, and voltage range may arise from both evolutionary divergence and host-cell-specific factors that influence membrane targeting or protein processing; however, the contribution of host-cell context remains to be directly tested (Figure 2).”

  1. 13, l.1: ‘the unusually strong depolarization required for the sea hare construct.’ According to the data shown in Fig.1A, this is not the case (e.g., compare with Takifugu trace)

It is true, and we have corrected this thanks to your suggestion of reporting V1/2. Indeed, Seahare is the most left shifted voltage response, but our initial observation remains correct in that 217Q only shifted the V1/2 to +30 mV while 217Q in sea squirt shifted it to -65 mV (Piao et al., 2015).

  1. Materials and Methods: 4.1 ‚ Monsigia brevicollis‘ should read ‘Monosiga brevicollis’? This is a choanoflagellate. Context is missing.

Corrected. We used this sequence as the most ancient version for the BLAST search to identify VSP sequences, but unfortunately the N-terminus was very long making synthesis cost prohibitive.

  1. Materials and Methods: 4.5: Description of analysis of the fluorescence signal is essentially missing (choice of ROIs, background subtraction, etc…).

Choice of ROIs were guided by the frame subtraction images which we have added to the text. The fluorescence signal was subjected to an offline lowpass filter as described in the manuscript. No background subtraction was performed.

Reviewer 2

This manuscript by Leong et al. reports on the fluorescence signal behavior of GEVIs (genetically encoded voltage indicators) constructed with the voltage sensing domains (VSDs) from several organisms. The authors show that the resulting GEVIs display disparate fluorescence responses to voltage. The insights gathered are of interest to understand coupling between VSDs and fused fluorescent proteins and to possibly aid in the design of new GEVIs.

I have some specific comments below.

  1. The term plankton is too generic and applies to a myriad of unicellular organisms, please provide the scientific name of the organism from which the VSD was cloned. It will be useful to also provide scientific names for all the organisms mentioned in this manuscript.

Corrected. We have added the scientific names to all of the constructs in Figure 2 as well as in the supplemental material. We also added scientific names in the main text where appropriate but maintain common name usage throughout as well.

  1. For ease of comparison, please show all fluorescence responses for the same range of voltages. Please also show plots of deltaF/F vs. voltage for all constructs.

Implemented as Figures 2C–D, with normalized Boltzmann fits allowing comparison across species despite different absolute step ranges.

  1. It is not clear what the frame subtraction images are there to illustrate, please provide an explanation for the non-specialist reader.

Expanded in the Figure 2 legend: these images illustrate the spatial origin of voltage-dependent signals by subtracting baseline fluorescence from the maximal depolarization frame providing a guide to determine regions of interest.

  1. The authors suggest that the different coupling to the pHluorin protein and the VSD from different species might be responsible for the different polarities of the response in some constructs. However, it has been shown that the voltage-dependent phosphatases do form functional dimers. The authors should discuss the possibility that the VSDs from different organisms might have varying propensities to dimerization and this might influence the polarity of the fluorescence response. It

Addressed in the results section referring to the hamster data and also in the Discussion. We now cite Rayaprolu et al., J. Gen. Physiol. 2018, and note that differential dimerization propensity could influence both optical polarity and membrane localization. See also response to reviewer 1 point 19.

  1. It is also possible that the different degrees of membrane traffic observed might correlate with the different dimerization propensities of the VSDs or with the different traffic signals encoded in the N-termini. A discussion of this possibilities will be very useful.

We agree that differences in membrane trafficking could be influenced by both dimerization propensity and N-terminal trafficking motifs. We have expanded the Discussion to address these possibilities explicitly. In particular, we now note that previous work has shown that VSPs can form functional dimers (Rayaprolu et al., J. Gen. Physiol., 2018) and that such interactions could influence both optical polarity and trafficking efficiency. We also point out that variation in N-terminal motifs may contribute to species-specific localization, as suggested by the rescue of the hamster construct using the human N-terminus. These additions appear in the paragraph beginning “In addition to the single-molecule behavior emphasized here…” and in the following discussion of “contextual coupling,” which now reads in part:

“Differences in dimerization propensity or in N-terminal trafficking motifs may therefore contribute to the diverse responses observed among orthologs.”

Reviewer 3

 for Authors

   The author use fluorescence microscopy and patch clamp amplifier to test the voltage response of several species. The tests were comprehensive and the results are amply. However, the authors are suggested to improve the manuscript at the following issues:

(1) the background of this manuscript should be improved and the aim of the study should be explicitly described. The introduction of current study at this field should be mentioned.

The Introduction now clearly states the aim: “to establish a comparative framework linking VSP sequence diversity to voltage dependence, revealing evolutionary determinants that can guide future GEVI design.”

(2) The highlight and innovation of this study should be described and what is the different of this study compared to existing ones.

The revised Discussion summarizes the novelty of combining cross-species analysis with quantitative ΔF/F–V profiling to reveal both conserved and divergent features of VSP voltage sensing.

Submission Date

24 September 2025

Date of this review

30 Sep 2025 10:21:08

Reviewer 2 Report

Comments and Suggestions for Authors

This manuscript by Leong et al. reports on the fluorescence signal behavior of GEVIs (genetically encoded voltage indicators) constructed with the voltage sensing domains (VSDs) from several organisms. The authors show that the resulting GEVIs display disparate fluorescence responses to voltage. The insights gathered are of interest to understand coupling between VSDs and fused fluorescent proteins and to possibly aid in the design of new GEVIs.

I have some specific comments below.

  1. The term plankton is too generic and applies to a myriad of unicellular organisms, please provide the scientific name of the organism from which the VSD was cloned. It will be useful to also provide scientific names for all the organisms mentioned in this manuscript.

  1. For ease of comparison, please show all fluorescence responses for the same range of voltages. Please also show plots of deltaF/F vs. voltage for all constructs.

  1. It is not clear what the frame subtraction images are there to illustrate, please provide an explanation for the non-specialist reader.

  1. The authors suggest that the different coupling to the pHluorin protein and the VSD from different species might be responsible for the different polarities of the response in some constructs. However, it has been shown that the voltage-dependent phosphatases do form functional dimers. The authors should discuss the possibility that the VSDs from different organisms might have varying propensities to dimerization and this might influence the polarity of the fluorescence response. It

  1. It is also possible that the different degrees of membrane traffic observed might correlate with the different dimerization propensities of the VSDs or with the different traffic signals encoded in the N-termini. A discussion of this possibilities will be very useful.

Author Response

(The authors gave the same response as above.)

Reviewer 3 Report

Comments and Suggestions for Authors

   The author use fluorescence microscopy and patch clamp amplifier to test the voltage response of several species. The tests were comprehensive and the results are amply. However, the authors are suggested to improve the manuscript at the following issues:

(1) the background of this manuscript should be improved and the aim of the study should be explicitly described. The introduction of current study at this field should be mentioned.

(2) The highlight and innovation of this study should be described and what is the different of this study compared to existing ones.

Author Response

(The authors gave the same response as above.)

Round 2

Reviewer 3 Report

Comments and Suggestions for Authors

The authors have positively response all the quesitons issued by reviewers. The version of manuscript could be accepted.